# Tissue-specific *in vivo* transformation of plasmid DNA in Neotropical tadpoles using electroporation

Jesse Delia[1¤]*, Maiah Gaines-Richardson[1], Sarah C. Ludington[1], Najva Akbari[1], Cooper Vasek[2], Daniel Shaykevich[1], Lauren A. O'Connell[1]*

1 Department of Biology, Stanford University, Stanford, CA, United States of America, 2 Department of Biology, De Anza College, Cupertino, CA, United States of America

¤ Current address: Division of Vertebrate Zoology, American Museum of Natural History, New York, NY, United States of America

* jdelia@amnh.org (JD); loconnel@stanford.edu (LAO)

**Data Availability Statement:** The data and code are uploaded to the supporting information S1 Data as a zip folder.

## Abstract

Electroporation is an increasingly common technique used for exogenous gene expression in live animals, but protocols are largely limited to traditional laboratory organisms. The goal of this protocol is to test *in vivo* electroporation techniques in a diverse array of tadpole species. We explore electroporation efficiency in tissue-specific cells of five species from across three families of tropical frogs: poison frogs (Dendrobatidae), cryptic forest/poison frogs (Aromobatidae), and glassfrogs (Centrolenidae). These species are well known for their diverse social behaviors and intriguing physiologies that coordinate chemical defenses, aposematism, and/or tissue transparency. Specifically, we examine the effects of electrical pulse and injection parameters on species- and tissue-specific transfection of plasmid DNA in tadpoles. After electroporation of a plasmid encoding green fluorescent protein (GFP), we found strong GFP fluorescence within brain and muscle cells that increased with the amount of DNA injected and electrical pulse number. We discuss species-related challenges, troubleshooting, and outline ideas for improvement. Extending *in vivo* electroporation to non-model amphibian species could provide new opportunities for exploring topics in genetics, behavior, and organismal biology.

## Introduction

The ability to transfer macromolecules into cells in intact animals offers exciting opportunities to investigate cellular morphology, development, and function *in vivo*. Electroporation provides a versatile and efficient method to transfect cells in targeted tissues [1–4]. This technique uses standard electrophysiological equipment to deliver a series of short electrical pulses to a targeted area of the animal, which temporarily increases cell-membrane permeability and transports charged macromolecules into cells via electrophoresis [2, 5, 6]. Electroporation has been used for a variety of objectives, from visualizing cells by transfecting plasmid encoded markers, to functional tests of regulation and foreign gene expression by delivering a range of

**Funding:** This work is supported by a grant from the National Science Foundation (www.nsf.org; IOS-1827333). and New York Stem Cell Foundation (https://nyscf.org/; NYSCF-R-NI58) to LAO. During analyses and drafting, JD was supported by a Gerstner Scholars Fellowship provided by the Gerstner Family Foundation and the Richard Gilder Graduate School at the American Museum of Natural History (https://gerstner.org/; www.amnh.org/research/richard-gilder-graduate-school/). DS is supported by a NSF Graduate Research Fellowship (www.nsf.org; DGE-1656518). LAO is a New York Stem Cell Investigator. The funders had no role in study design, data collection and analysis, decision to publish, or preparation of the manuscript.

**Competing interests:** The authors have declared that no competing interests exist.

macromolecules (e.g., DNA, RNA, proteins, and morpholinos) [7–11]. Importantly, transfection efficiency is influenced by several conditions including electrical pulse parameters, cell-membrane composition, cell orientation and tissue type, as well as species differences in biological parameters [5–12]. Therefore, exploring the electroporation parameters that transfer bioactive compounds without causing cell death is important for building capacity for functional tests in diverse organisms [4, 13].

Among amphibians, most electroporation protocols were developed for model species, such as *Xenopus*, axolotls, and newts ([14–16]; but see [17]). Indeed, there are many protocols detailing methods optimized for *Xenopus* tadpoles that include electroporation of brains [3, 8, 18–22], eyes [23], and tails [16, 24]. Here, we develop electroporation protocols for tadpoles of five species of Neotropical poison frogs (Dendrobatidae), cryptic forest/poison frogs (Aromobatidae), and glassfrogs (Centrolenidae). Tadpoles from these groups exhibit a diversity of phenotypes, involving parent–offspring interactions and communication, sociality, and tissue transparency (Fig 1, [25–29]). Extending protocols for *in vivo* gene transfer beyond *Xenopus* tadpoles could provide new opportunities to explore biological diversity and address questions in organismal biology, neuroscience and physiology within ecological and evolutionary frameworks.

Here, we provide electroporation protocols for brain cells and muscle fibers in tadpoles of the poison frogs *Dendrobates tinctorius*, *Ranitomeya imitator*, *Ranitomeya variabilis*, the cryptic poison frog *Allobates femoralis*, and the glassfrog *Hyalinobatrachium fleischmanni*. By adapting techniques developed for *Xenopus* tadpoles, we explored the effects of electroporation pulse parameter and DNA injection volume on the transfection efficiency of a plasmid encoding the gene for green fluorescent protein (GFP). We report electroporation efficiency across species and tissue types and discuss instances of common problems, troubleshooting, and prospects.

## Materials and methods

The protocol described in this peer-reviewed article is published on protocols.io, https://dx.doi.org/10.17504/protocols.io.8epv5jjq4l1b/v1 and is included for printing as S1 File with this article.

## Ethics declarations

All animal procedures were approved by the Institutional Animal Care and Use Committee at Stanford University (protocol number #33016). An MS222 bath was used as an anesthesia prior to electroporation, imaging, and euthanasia of all animals.

## Animals

We worked with captive-bred tadpoles of the glassfrog *Hyalinobatrachium fleischmanni* (Centrolenidae), the cryptic poison frog *Allobates femoralis* (Aromobatidae), and three poison frogs *Dendrobates tinctorius*, *Ranitomeya imitator*, and *Ranitomeya variabilis* (Dendrobatidae) (Fig 1). Tadpoles of the African clawed frog *Xenopus laevis* (Pipidae) were purchased from a commercial supplier (Nasco, Fort Atkinson, WI, USA). The species studied here lay larger ($\geq$ 2 mm) but fewer eggs at oviposition compared to *Xenopus* (~1.2 mm). The non-model embryos also hatch larger and at more developed stages, although *X. laevis* can grow to a larger size at metamorphosis [37–41]. Many of these Neotropical species also exhibit considerable plasticity in hatching age, being capable of hatching at different development stages and hatching in response to environmental cues and parents [28, 42–44]. Differences in species sampling for each protocol were due to availability tadpoles during protocol development.

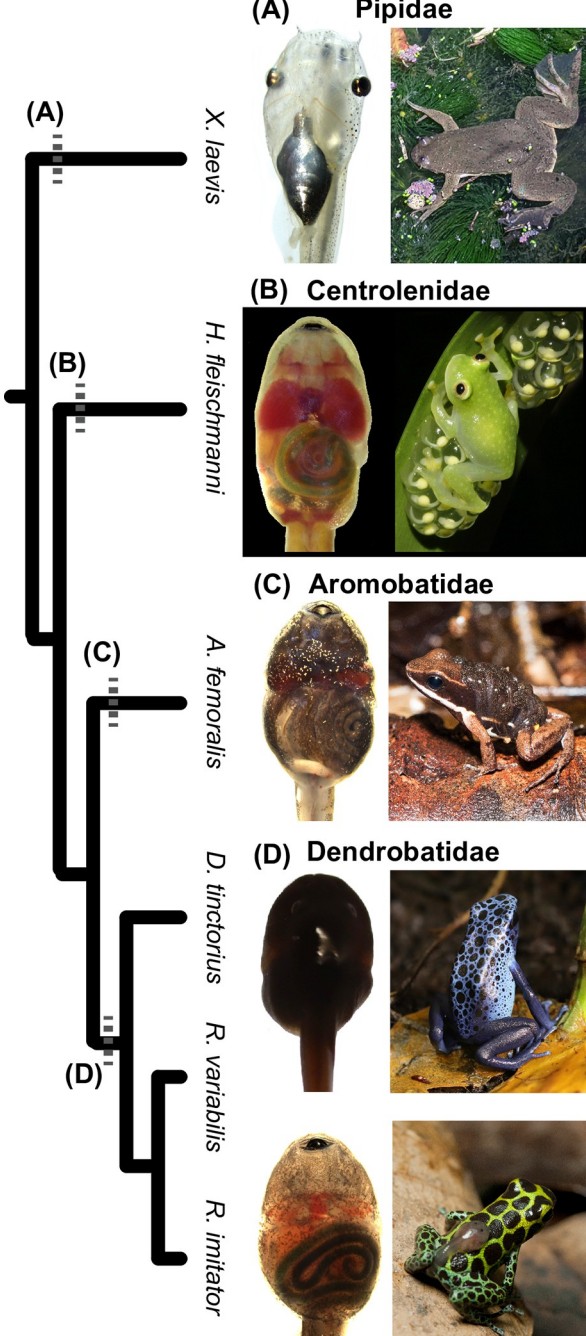

**Fig 1. Species relationships and phenotypes of the study species.** (**A**) The African clawed frog *Xenopus laevis* (Pipidae) is a model organism for laboratory studies and for developing *in vivo* techniques for testing gene function. Images by JD, the adult was photographed near Hluhluwe, KwaZulu-Natal, South Africa. (**B**) The glassfrog *Hyalinobatrachium fleischmanni* (Centrolenidae) exhibits transparent tissues, through which many organs are visible during embryonic, tadpole and adult life-stages. Fathers provide parental care to developing eggs and embryos can time hatching in response to parenting [28–31]. Images by JD, the adult was photographed near San Gabriel Mixtepec, Oaxaca, Mexico. (**C**) *Allobates femoralis* is a cryptic poison frog (Aromobatidae) whose adult behavior is well studied in the context of spatial cognition and parental behavior [32–35]. Images by DS (tadpole) and Andrius Pašukonis (adult from Nouragues Nature Reserve, French Guiana). (**D**) Poison frogs of the Dendrobatidae are well known for their aposematic coloration, chemical defenses, and diverse social behaviors [25–27, 36]. Images are of *Dendrobates tinctorius* by DS (captive animals) and *Ranitomeya imitator*, by DS (tadpole) and Evan Twomey (adult from near San Jose, San Martin, Peru).

For all experiments, we anesthetized tadpoles by bathing them in 0.03% 3-aminobenzoic acid ethyl ester (MS222, Millipore Sigma, St Louis, MO, USA) buffered with 0.1% sodium bicarbonate prior to the electroporation procedure, then again prior to live imaging. Tadpoles were euthanized via immersion in a lethal dose of buffered 0.15% MS222 before brain dissection. Electroporation procedures were conducted on tadpoles in stages ranging between 25–29 Gosner [45] stages or 44–49 [37] stages for *Xenopus*. Following injection and electroporation, all tadpoles were transferred to fresh tadpole water (DI with osmolyte drops, Josh's Frogs R/O rx, Josh's Frogs, Owosso, MI, USA) for up to three hours for recovery and then moved to their home aquaria.

## *In vivo* microinjection of plasmid DNA

Glass capillaries (3.5-inch length, 0.02-inch width: Drummond Scientific) were pulled (Sutter Instrument Co P-97) and the tip was broken under a dissecting scope using forceps. We injected a solution of 0.25–0.27 μg/μl of purified plasmid DNA (pCMV-GFP, Addgene #11153) mixed with 0.01% Fast Green using a Nanoject II injector (Drummond Scientific). Injection volumes are listed below with the corresponding protocol. We used plasmid concentrations based on that reported for *X. laevis* [3, 16, 46].

## Practice sessions

Prior to formal experiments and data collection, we conducted practice trials to learn how to target specific tissues and evaluate species-specific ways to establish good electrode contact. These sessions were conducted for each species and each tissue type until we observed consistent transfection within replicate groups of tadpoles. Once successful and consistent transfection was confirmed for each species and tissue type, we initiated a formal experiment testing each protocol and collected the resulting quantitative data for statistical analyses.

## Muscle-cell electroporation

We tested whether we could label myocytes and experimentally alter the number of cells transfected in tadpole tails by modifying protocols developed for *Xenopus* muscle tissue [16, 46]. Briefly, we built an electroporation setup similar to that used by Mochii and Taniguchi (2009), using platinum foil for electrodes soldered with electric leads to wire them to the stimulator (Fig 2). The anode (~ 5 x 8 mm) was mounted on the end of a hard plastic tube (~1 cm diameter), and embedded flat in a platform of molding clay fixed on top of a Petri dish. The clay platform allowed us to quickly mold a size-specific depression in the clay around the anode, such that the anesthetized tadpole lay flat on its side with its head in the depression and a ~5 mm section of its tail across the anode plate. The cathode (~ 5 x 8 mm) was fixed to the end of a plastic tube (~1 cm diameter) and mounted to a micromanipulator. The tip of the cathode was bent such that a ~5 mm section could be pressed flat on the animal's tail directly above the anode—these probes intersected the animal's tail perpendicularly. We mounted a Nanoject injector on a second micromanipulator opposite to the electroporation setup with the two instruments placed on either side of a dissecting scope. After covering the platform with a moistened Kimwipe, we positioned the anesthetized tadpole on the platform. During the procedure, we injected the plasmid DNA into a myomere on the tail, and then delivered a series of electrical pulses to the injection area 5–10 s after the injection. The animal was transferred to fresh tadpole water to recover immediately following electroporation.

We conducted a preliminary experiment using *X. laevis* tadpoles to determine the range of electroporation parameters that could be applied to glassfrog and poison-frog tadpoles with low mortality. Using a group of 100 *X. laevis* tadpoles, we compared two treatments for each of

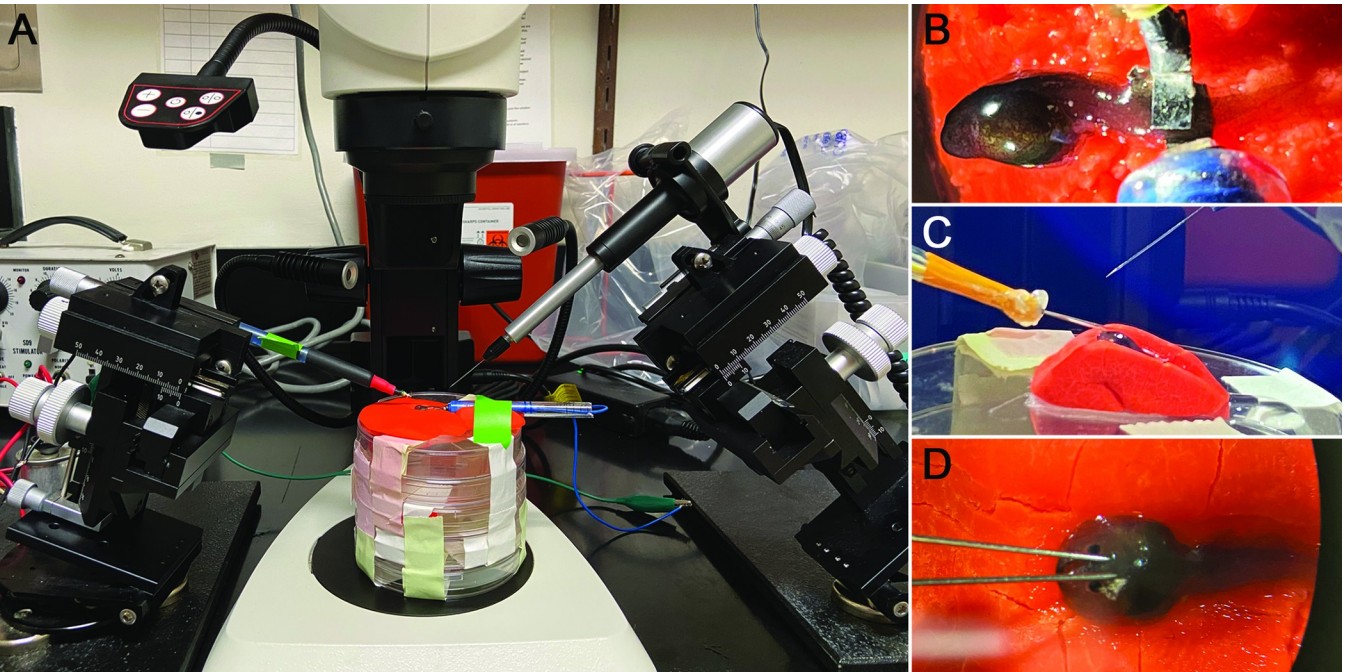

**Fig 2. Electroporation set-up. (A)** We placed the tadpole on a platform of modeling clay and positioned it under a dissecting scope to target the tissue for injection and electroporation. We positioned micromanipulators on both sides of the platform, mounted with either the electrode/s (L) or the microinjector (R). **(B)** Muscle electroporation in a *Ranitomeya imitator* tadpole. **(C)** Brain electroporation in a *Ranitomeya imitator* tadpole. Note that the clay has been molded to counteract the pressure and maintain head shape during injection and electroporation. **(D)** In tadpoles that are soft and globular, platinum needles are recommended instead of platinum foil sheets.

the following parameters (10 tadpoles per group): 82.4 vs 128.8 nl of plasmid DNA solution, 8 vs 16 pulses, 30 vs 50 volts, and a control group of injection-only tadpoles (82.4 nl). To manipulate DNA injection volumes, we delivered two injections per treatment level (of 41.2 nl vs 64.4 nl) spaced at 5–10 s intervals using the slow injection setting. We delivered double square pulses lasting 5 ms in duration spaced at 5 ms intervals, with each set of double pulses delivered at 1 s intervals using a Grass SD9 stimulator. We quantified mortality during the three days following the experiment and, on the third day, counted the numbers of GFP-positive myocytes by imaging the tadpoles under a fluorescent dissecting microscope with a GFP filter (Leica M165 FC) mounted with a camera (Leica DFC7000 T). We identified myocytes in *Xenopus* and the other species (below) based on their morphology using GFP-labeling and fluorescence microscopy. Myocytes are large cells that exhibit a distinct cylindrical morphology and are organized in parallel bundles within myomeres in tadpole tails [16, 24, 47, 48] (Fig 3). We counted myocytes directly under a dissecting (fluorescent) microscope, where we could quickly adjust light levels, the animal's position, and use forceps and probes to carefully examine the tail muscle to better delimit the edges of individual myocytes.

Next, we conducted experiments to generalize this method across *H. fleischmanni* ($n = 13$), *R. imitator* ($n = 20$), and *R. variabilis* ($n = 20$), in addition to repeating these parameters in *X. laevis* ($n = 20$). We found that DNA injection volume accounted for the largest proportion of variance in our statistical model of myocyte transfection in *X. laevis* and that the electroporation parameters of 30 volts and 8 pulses (4 double pulses) had low mortality (see results section). Therefore, we manipulated the injection volume (82.4 vs 128.8 nl of plasmid solution) and held all other electroporation parameters constant at 30 volts of 4 double pulses, each pulse 5 ms in duration spaced at 5 ms intervals. We quantified mortality for three days

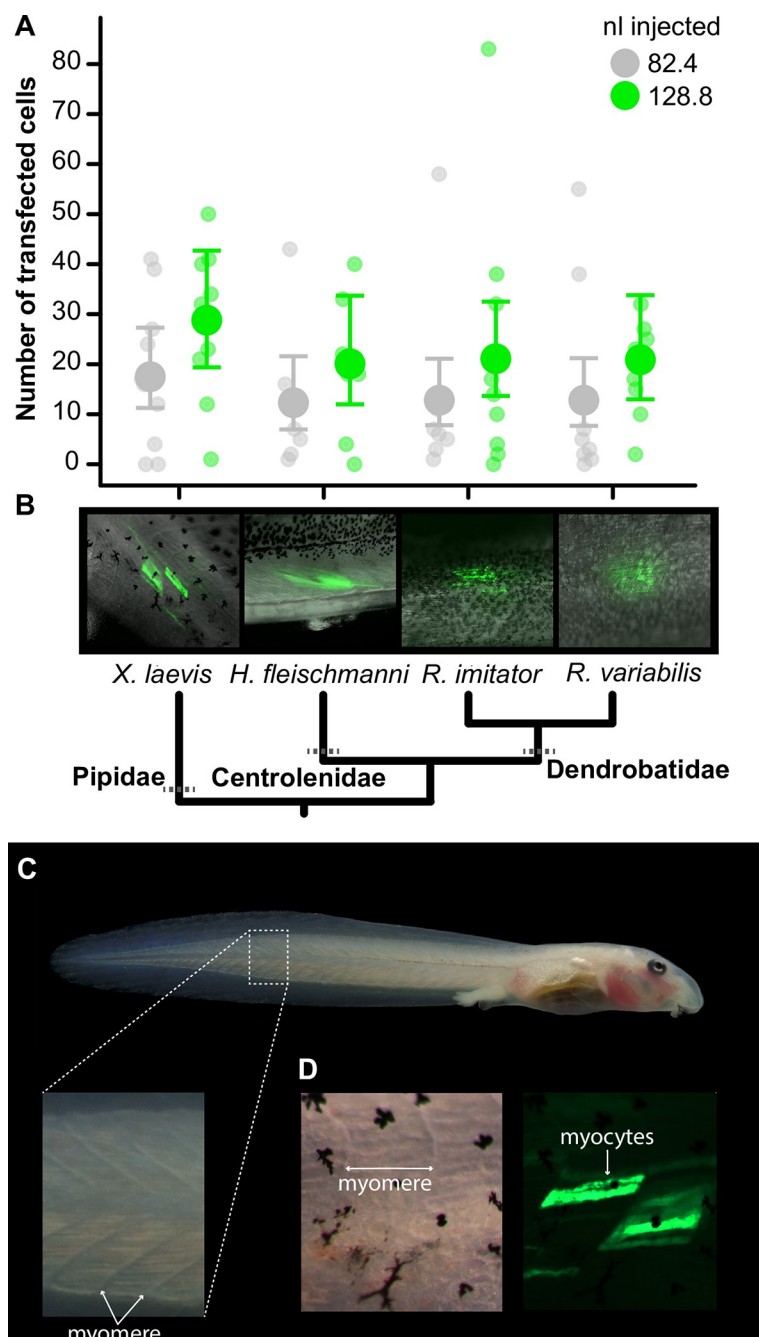

**Fig 3. GFP expression in muscle cells in tadpole tails. (A)** The number of transfected myocytes increased with the volume of plasmid DNA across all 4 species. Smaller circles show individual data points, larger circles and error bars show mean ± 95% CI (estimated from a quasi-Poisson GLMM). Datapoints are jittered to show stacked points. **(B)** Images of live tadpoles showing GFP-positive myocytes (green; from fluorescence microscopy) in their tail myomeres. Note that specie's differences in the extent of skin pigmentation and muscle-tissue opacity made counting cells difficult from images alone (i.e., adjusting exposure to capture specific cells can overexpose others and vice-versa depending on cell depth, position, and expression levels). Therefore, we counted myocytes directly under a dissecting (fluorescent) microscope, where we could adjust light levels, the animal's position, and use forceps and probes to delimit the edges of individual myocytes at different depths. **(C)** An image showing the targeted region of muscle tissue in a tadpole's tail. Inset: myomeres are blocks of muscle tissue that are arranged in sequence across the tadpole's tail. Note the individual myocytes arranged in parallel within myomeres, which are large cylindrical cells. **(D)** Brightfield (left) and fluorescent (right) images showing GFP-labeled myocytes within and delimited by the boarders of the myomeres.

following the experiment and counted GFP-positive myocytes *in vivo* using fluorescence microscopy (described above). Cell counts were likely influenced by species' differences in the extent of skin pigments and muscle-tissue opacity, with the platanna and *Hyalinobatrachium* glassfrogs being highly transparent, and the poison frogs having more pigmented skin (see images in Fig 3). Importantly we conducted a within-species experimental design and combined mixed-modeling to compare within species' effects. Our statistical model did not detect species-level or interaction effects (e.g., similar effect sizes across species, see results below).

## Brain-cell electroporation

We tested whether we could label brain cells of Neotropical tadpoles through electroporation of plasmid DNA. We started by testing protocols developed for *Xenopus* tadpole brains [3, 9]. For this setup, we soldered two ~1mm square platinum plates to separate wires and positioned them at the end of a single hard-plastic tube (~1 cm diameter), such that the plates were parallel and separated by ~1 mm (we made slight changes to this spacing as needed using dissecting forceps). We constructed a platform out of molding clay on top of a Petri dish. A body-sized depression was made on the mound edge to fit each tadpole, and each animal's head was draped slightly over the edge such that the probe would contact the brain area almost perpendicularly. This design was important to maintain pressure, body shape, and accuracy for both injections and electrode contact, as the Neotropical tadpole species used here are soft and deform with compression without proper support. We placed the platform underneath a dissection microscope with the electrode rod connected to a micromanipulator on one side and the Nanoject injector on a second manipulator on the other. After covering the platform with a moistened Kimwipe, we positioned the anesthetized tadpole on the platform. During the procedure, we injected plasmid DNA into the brain ventricle and then delivered a series of electrical pulses to the injection area 5–10 s after the injection. The animal was transferred to fresh tadpole water to recover immediately following electroporation.

Based on preliminary sessions and results from others [9], we decided to manipulate pulse number and keep all other parameters constant. Tadpoles (*A. femoralis* n = 28, *D. tinctorius* n = 32, *H. fleischmanni* n = 42, and *R. imitator* n = 62) received either 4 or 10 square (single) pulses of 30 volts lasting 50 ms in duration and spaced at ~ 1 s intervals. We delivered a total 193.2 nl of DNA plasmid solution into the ventricle with three injections of 64.4 nl spaced at 5–10 s using the slow setting of the injector. We quantified mortality for three days following the experiment. On the third day, we imaged live tadpole heads (using the fluorescence microscope setup described above), then euthanized them and immediately collected their brains. Tadpole brains were fixed in 4% paraformaldehyde for 24 h, washed in 1x phosphate-buffered saline, and then the dorsal side of whole-brains were photographed using the fluorescence microscope setup described above. These images account for the signal that can be detected only in dorsal view, and GFP signal was likely influenced by species differences in brain size, opacity, and melanin content. However, this rough estimate is sufficient to confirm successful labeling and examine for relative differences in GFP-signal area.

Images were processed and measured in ImageJ v 1.53a [49]. For each brain, we built a stack using a brightfield overlay and a corresponding fluorescent GFP image—the fluorescent image was median filtered to reduce noise (using the default setting of 2 pixels). We measured brain area from the optic tectum to the forebrain in the brightfield image. To select this region and facilitate segmenting, we first created a straight line across the brain at the posterior edge of the optic tectum, erased nerves and connective tissues that remained on the brain during imaging, selected this region using the magic wand tool, and measured area as the number of pixels. We then converted the GFP image to 8-bit and conducted Otsu thresholding [50] to

segment regions with GFP signal within the same selected brain region using automated settings in ImageJ. For analyses, we calculated the proportional area of brain tissue with GFP signal as the area with GFP signal divided by the total brain area (in pixels, in dorsal view).

We examined whether we could visualize GFP-labeled brain cells *in vivo* by using two-photon imaging (Prairie Ultima IV Two-photon microscope) of a *R. imitator* tadpole. After the animals were anesthetized, they were embedded in 2% agarose solution for mechanical stabilization. Animals were centered in the field of view using a low magnification objective lens to ensure the brain is within the viewing range of the high magnification two-photon objective lens. Images were collected with 920 nm excitation wavelength to capture GFP-signal distribution.

## Data analysis

We conducted statistical analyses in R v 3.6.1 [51]. For across species experiments, we used a generalized linear mixed modeling approach (GLMM) in the package glmmTMB [52]. This approach allowed us to account for random variance attributed to different experimental sessions while modeling the approximate error distribution that best handled each dataset. For these models, we tested the main effects of experimental treatment (either DNA plasmid volume, or pulse number), species, and their interaction. To analyze the preliminary experiment using *X. laevis* tadpoles—evaluating key parameters for myocyte transfection—we computed a GLM using the package stats, as this experiment was conducted over only two experimental sessions. We selected the appropriate error distribution for each model by first testing residual model fit using a series of diagnostic plots and, for GLMMs, using plots and test functions in the DHARMa package (version 0.3.3.0, [53]). For statistical inference of the fixed effects, we calculated *P* values using likelihood ratio tests (LRT) comparing increasingly simplified models. When we did not detect a significant interaction effect, we dropped it from the model and reran it to test and report the main fixed effects. We estimated effect sizes, group means, and 95% CI directly from all models and, when necessary, conducted post-hoc comparison (Tukey method) in the package emmeans (version 1.4.5, [54]).

To determine whether our experimental treatments influenced the likelihood of tadpole mortality, we modeled the binary outcome of alive/dead using GLMMs with a binomial error distribution and logit-link function. We modeled tadpole fates for only the brain-cell protocol (4 species), as only 1 in 73 tadpoles died during the across species experiment of muscle-cell electroporation (4 species).

For the myocyte dataset generated from *X. laevis*, our goal was to determine the range of key parameters to test in glassfrog and poison frog tadpoles. We modeled the number of transfected myocytes in *X. laevis* using a quasi-Poisson GLM with a log-link function ($n = 87$) and examined the fix effects of pulse number (8 vs 16), voltage (30 vs 50), the volume of DNA solution injected (82.4 vs 128.8 nl), and their interactions. Next, we tested whether DNA injection volume (82.4 vs 128.8 nl), species, and their interaction impacted the number of detectable GFP-positive myocytes using the cross-species dataset ($n = 68$). For both datasets, initial models using Poisson error distributions were overdispersed. Therefore, we modeled the number of transfected cells using a quasi-Poisson distribution (family 'nbinom2') and a log-link function, which better accounts for overdispersed count data [55].

For the brain-cell dataset, we tested whether pulse number, species, and their interaction influence successful transfection by modeling the proportion of brain area with GFP-positive cells using a GLMM with a beta-binomial error and logit-link function. These models are appropriate for bounded data and account for overdispersion typical of clumped proportional estimates [56].

## Expected results

### Exogenous gene expression in muscle tissue

We first explored electroporation parameters using *X. laevis* tadpoles and found that increasing the volume of DNA plasmid solution (82.4 vs 128.8 nl) significantly increased the number of transfected myocytes among the surviving tadpoles (87 of 100 individuals) (quasi-Poisson GLM: $\chi^2 = 26.43$, $p < 2.72e-07$). We did not detect pulse-number effects, voltage effects (pulse number $\chi^2 = 0.24$, $p = 0.62$; voltage $\chi^2 = 1.99$, $p = 0.15$), nor interaction effects in the initial model (all interactions $p > 0.05$).

Next, we determined if these parameters generalized across tadpole species and found that the volume of plasmid DNA injected into tail myomeres significantly increased the number of transfected myocytes (quasi-Poisson GLMM: $\chi^2 = 5.7$, $p = 0.017$, $n = 68$, Fig 3). We did not detect species-level effects ($\chi^2 = 2.17$, $p = 0.54$) or interaction effects (in the initial model; $\chi^2 = 1.09$, $p = 0.77$), indicating that this protocol worked equally well across the major tadpole families tested here. Furthermore, tadpole mortality was very low during our experiment, occurring in only 1 of 73 individuals (in *R. variabilis*); tail loss occurred in 4 individuals at the site of electroporation.

### Exogenous gene expression in the brain

We adapted protocols originally designed for *X. laevis* tadpoles [3, 9] and successfully expressed an exogenous gene (GFP) from a plasmid into brain cells of tadpoles from the Aromobatidae, Centrolenidae, and Dendrobatidae families (Fig 4). Mortality was relatively low during this experiment, with species-level averages ranging from 0% to 33% mortality (depending on the treatment level). We found that increasing pulse number (4 vs 10) significantly increased mortality (binomial GLMM: $\chi^2 = 23.5$, $p = 1.26e-06$, $n = 164$, Fig 4A). We did not detect species-level effects ($\chi^2 = 0.8$, $p = 0.85$) or interaction effects (in the initial model; $\chi^2 = 3.24$, $p = 0.35$).

Next, we found that increasing pulse number (4 vs 10) significantly increased the proportion of brain area with GFP-positive cells across all species (beta-binomial GLMM: $\chi^2 = 20.77$, $p = 5.17e-06$, $n = 97$, Fig 4B). We also detected species-level effects ($\chi^2 = 16.07$, $p = 0.001$). However, they did not interact with the experimental treatment (in the initial model, $\chi^2 = 3.41$, $p = 0.33$), indicating that transfection success was influenced by the treatment in all species. Differences between species appeared to be due to low levels of transfection in dendrobatids, with both *D. tinctorius* and *R. imitator* exhibiting the lowest level of transfection and the smallest treatment-effect sizes compared to the centrolenid *H. fleischmanni* (post-hoc comparison of species-level contrasts with *H. fleischmanni*: $p = 0.0009$ and $0.0003$, respectively). The aromobatid *A. femoralis* appeared to fall out intermediate between the glassfrog ($p = 0.64$) and the dendrobatids (contrast with *D. tinctorius* $p = 0.084$, and with *R. imitator* $0.047$). Finally, we were able to capture GFP signal in the brain cells of a live *R. imitator* tadpole using two-photon microscopy (Fig 4D).

There are several steps investigators could pursue to optimize electroporation in their species of interest. Other pulse settings, not explored here, were shown to be effective in *Xenopus* brains (e.g., pulses with exponential decay, [3]). Experimenters are encouraged to explore additional pulse types and develop electrodes and/or platforms that improve contact with the brain of the species being manipulated, as tadpoles shape and size varies across species. In addition, this electroporation protocol can be further optimized by users to induce cellular uptake of various molecules, as has been done in *Xenopus* tadpoles, including morpholinos [57, 58], fluorescent dyes [59], and or single-cell electroporation methods [18, 19, 22].

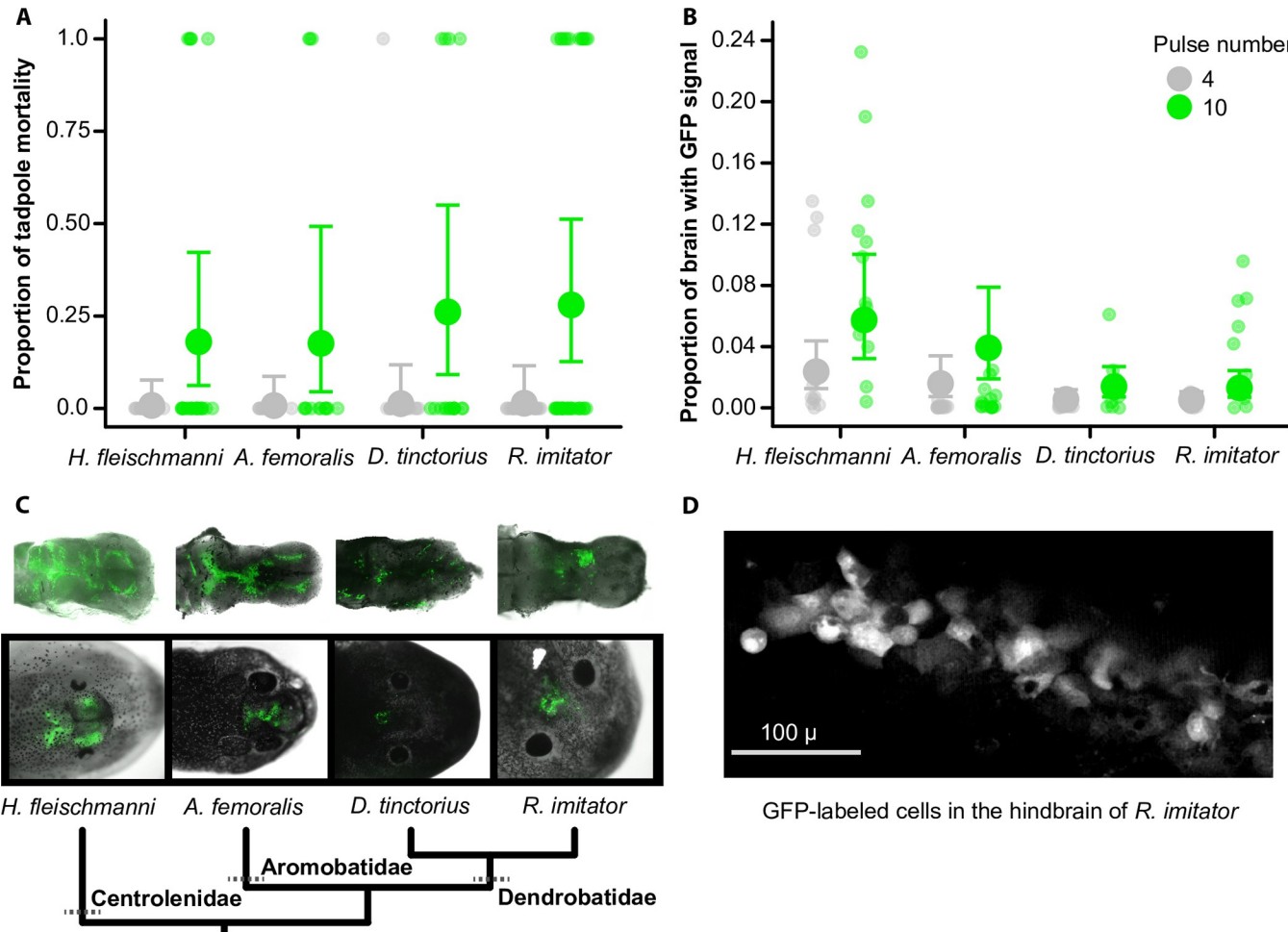

**Fig 4. Tadpole mortality and GFP expression in brain cells. (A)** Tadpole mortality increased with pulse number during brain electroporation. Smaller circles show tadpoles that survived (0) or died (1), larger circles and error bars show the mean ± 95% CI of the proportion of tadpoles that died (estimated from a binomial GLMM). **(B)** The proportional area of transfected brain cells increased with pulse number across all species. However, the magnitude of treatment effects varied by species. Smaller circles show the proportional area of individual brains with GFP-positive cells, larger circles and error bars show the mean ± 95% CI (estimated from a beta-binomial GLMM). In both plots, data points are jittered to show stacked points. **(C)** Images of dissected brains (top row) and live tadpoles (bottom row) showing GFP-positive brain cells (green; from fluorescence microscopy). **(D)** An image from two-photon microscopy showing GFP-labeled cells *in vivo* within the hindbrain of an *R. imitator* tadpole.

We found that the shape of the electrodes likely impacts successful brain electroporation. For the two Dendrobatid species, inconsistent electrode contact contributed to low levels of brain-cell transfection, as we found it difficult to wrap the electrodes around the brain due to their globular and soft body-structure. When pressing the electrodes into their head, the brain shifts ventrally into the body, such that the electrode contact was primarily limited to the skin and muscle tissue. Moreover, skin pigmentations in these two species made it difficult to visually confirm electrode contact with the brain and there appeared to be a high occurrence of poor contact in our experiment. To account for the potential effects of electrode shape on plasmid uptake in brain cells, we redesigned the electrodes for a single tadpole species (*R. imitator*). We purchased disposable paired 13 mm subdermal needle electrodes to replace the foil-plate electrodes. The electrode wires were each run separately through a hard plastic tube, then mounted on a micromanipulator side-by-side with the needles ~1 mm apart and parallel to one another. The lighter weight of the needles reduced brain movement and thus improved

electrode contact with the brain. We also observed reduced skin burning using this new setup. Moreover, electroporation efficiency likely impacts the extent of mosaicism, as species with more globular shapes tend to have sparser and asymmetric uptake in the brain. Thus, we recommend optimizing electrode shape to account for species differences in tadpole shape.

## Supporting information

**S1 File. Step-by-step protocol, also available on protocols.io.**
(PDF)

**S1 Video. Protocol video.**
(MP4)

**S1 Data.**
(ZIP)

## Acknowledgments

We thank Eva Fischer for conducting preliminary trials of brain-cell electroporation in *Xenopus* tadpoles, which helped us in protocol development. We thank members of the O'Connell Lab for frog colony care, general experimental advice, and encouragement. We thank Lukas Weiss and Carlos Taboada for advice and ideas about troubleshooting methods. We thank Evan Twomey and Andrius Pašukonis for providing images in Fig 1. We also thank Hollis Cline and Caroline McKeown, who taught LAO how to electroporate *Xenopus* tadpoles. We acknowledge that this research was conducted at Stanford University, which is located on the ancestral and unceded land of the Muwekma Ohlone tribe.

## Author Contributions

**Conceptualization:** Jesse Delia, Lauren A. O'Connell.

**Data curation:** Jesse Delia.

**Formal analysis:** Jesse Delia.

**Funding acquisition:** Lauren A. O'Connell.

**Investigation:** Jesse Delia, Maiah Gaines-Richardson, Sarah C. Ludington, Najva Akbari, Cooper Vasek.

**Methodology:** Jesse Delia, Maiah Gaines-Richardson.

**Project administration:** Lauren A. O'Connell.

**Resources:** Lauren A. O'Connell.

**Supervision:** Lauren A. O'Connell.

**Validation:** Jesse Delia.

**Visualization:** Jesse Delia, Maiah Gaines-Richardson, Najva Akbari, Daniel Shaykevich.

**Writing – original draft:** Jesse Delia, Lauren A. O'Connell.

**Writing – review & editing:** Jesse Delia, Maiah Gaines-Richardson, Sarah C. Ludington, Daniel Shaykevich, Lauren A. O'Connell.

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
