## [Decision Letter · Decision Letter 0]

6 Mar 2023

PONE-D-23-02696Tissue-specific in vivo transformation of plasmid DNA in Neotropical tadpoles using electroporationPLOS ONE

Dear Dr. Delia,

Thank you for submitting your manuscript to PLOS ONE. After careful consideration, we feel that it has merit but does not fully meet PLOS ONE’s publication criteria as it currently stands. Therefore, we invite you to submit a revised version of the manuscript that addresses the points raised during the review process.

We look forward to receiving your revised manuscript.

Kind regards,

Michael Schubert

Academic Editor

PLOS ONE

Reviewers' comments:

Reviewer's Responses to Questions

**Comments to the Author**

1. Does the manuscript report a protocol which is of utility to the research community and adds value to the published literature?

Reviewer #1: Yes

Reviewer #2: Yes

2. Has the protocol been described in sufficient detail?

To answer this question, please click the link to protocols.io in the Materials and Methods section of the manuscript (if a link has been provided) or consult the step-by-step protocol in the Supporting Information files.

The step-by-step protocol should contain sufficient detail for another researcher to be able to reproduce all experiments and analyses.

Reviewer #1: Partly

Reviewer #2: Yes

3. Does the protocol describe a validated method?

Reviewer #1: Yes

Reviewer #2: Yes

4. If the manuscript contains new data, have the authors made this data fully available?

Reviewer #1: Yes

Reviewer #2: Yes

**5. Is the article presented in an intelligible fashion and written in standard English?**

Reviewer #1: Yes

Reviewer #2: Yes

6. Review Comments to the Author

Reviewer #1: In their Lab Protocol entitled "Tissue-specific in vivo transformation of plasmid DNA in Neotropical tadpoles using electroporation," Jesse Delia and collaborators successfully adapted the electroporation technique to a range of non-model species tadpoles.

The electroporation technique has been used to transfer charged macromolecules, including fluorescent dyes and DNA, into living cells of Xenopus tadpoles for decades. In the present Protocol, the authors show that this technique is also suitable for transferring plasmid DNA into living cells of tadpoles of other species.

Although it is not very surprising that the electroporation technique works on tadpoles of different species, I believe that the present Protocol will be well received in the amphibian community.

The Protocol is well-written and straightforward. It will encourage researchers working with different amphibian species to use this powerful technique.

I would have loved it if you had also tried electroporating single cells. In your experiments, you perform bulk electroporations. Labeling specific cell types or cells of interest will be impossible with your method.

You could be more specific when stating what type of experiments can be done or improved using the described electroporation technique. Many experiments that I have in mind would necessitate a more precise and controlled electroporation of specific cells/ cell types.

I only have a few questions and suggestions that the authors could consider taking into consideration when revising their Protocol. I hope this helps to make some points more precise.

Specific points

You solely report the successful electroporation of plasmid DNA into cells. Have you tried to electroporate fluorescent dyes, calcium indicators, etc.?

Some tadpole-specific electroporation Protocols are missing from your publication list.

While in Figure 2D, the position of the electrode on the tadpole is nicely visible, the other three subfigures of Figure 2 are not that helpful. The electrode position on the tadpoles is not clearly visible.

Is this a kinked tail of the tadpole sticking up in Figure 2B?

Figure 3: When looking at the pictures you provide in Figure 3B, I cannot imagine how to count single cells. What are the borders of a single myocyte? How can you even be sure that these are all myocytes?

It is difficult to understand how you calculated the proportional area of transfected brain cells.

You state that non-model species of tadpoles can survive higher voltage, pulse number, and pulse duration if compared to Xenopus. Where do you show this?

Line 260: …." using a using"….Please correct this part of the sentence.

Figure 4D: How can you be sure that the depicted cells are neurons? Where are these cells located in the brain?

Reviewer #2: The manuscript by Delia et al is focus on plasmid DNA electroporation in non-model frog at tadpole stage. This work is really interesting because is one of the first description of this technique on non-model frog. The video and protocol are really well do and appreciable. This study can be accepted after major modification.

Major comments

1- Xenopus is a classical frog model and it’s used as reference here to develop technics on non-model frog. It’s known that Xenopus Laevis embryos are commonly described as having a large size. Author can compare the size of non-model embryos at tadpole stage to Xenopus Laevis embryos ?

2 – Authors used five various non-model frog in this study. Authors used some species for myogenic experiment and other for neural tissue. Authors could explain that ?

3- In fig 3 authors described GFP expression in muscle cells. I’m not be able to see cells on picture and I’m not be able to localise fluorescent areas. Authors could add lower magnification to see where is located GFP on tadpole and higher magnification to see the counted cells.

4- It is known that embryonic electroporation can be give mosaicism especially in nervous system but also in myogenic tissue. Author can discuss this point and precise when they talk about nervous system if GFP is often found in the same area or in various neural area.

Minor point

1- In phylogenetic tree author show one photo for two different species concerning Aromobatidae and Dendrobatidae. Authors can give a photo of each species.

2- Line 260: removed one “a using”.

3- Some scale bars are missing for fig 3 and 4. Authors could add it ?

7. PLOS authors have the option to publish the peer review history of their article (what does this mean?). If published, this will include your full peer review and any attached files.

Reviewer #1: No

Reviewer #2: No

---

## [Author Response · Author response to Decision Letter 0]

20 Jun 2023

We have addressed all comments and concerns provided by the editor and two reviewers. Please see our cover letter and response to reviewers doc for details.

---

## [Decision Letter · Decision Letter 1]

18 Jul 2023

Tissue-specific in vivo transformation of plasmid DNA in Neotropical tadpoles using electroporation

PONE-D-23-02696R1

Dear Dr. Delia,

We’re pleased to inform you that your manuscript has been judged scientifically suitable for publication and will be formally accepted for publication once it meets all outstanding technical requirements.

Kind regards,

Michael Schubert

Academic Editor

PLOS ONE

Reviewers' comments:

Reviewer's Responses to Questions

**Comments to the Author**

1. Does the manuscript report a protocol which is of utility to the research community and adds value to the published literature?

Reviewer #2: Yes

2. Has the protocol been described in sufficient detail?

To answer this question, please click the link to protocols.io in the Materials and Methods section of the manuscript (if a link has been provided) or consult the step-by-step protocol in the Supporting Information files.

The step-by-step protocol should contain sufficient detail for another researcher to be able to reproduce all experiments and analyses.

Reviewer #2: Yes

3. Does the protocol describe a validated method?

Reviewer #2: Yes

4. If the manuscript contains new data, have the authors made this data fully available?

Reviewer #2: Yes

**5. Is the article presented in an intelligible fashion and written in standard English?**

Reviewer #2: Yes

6. Review Comments to the Author

Reviewer #2: Hi,

Thank you for yours answers. I'm satisfied.

Have a nice day,

7. PLOS authors have the option to publish the peer review history of their article (what does this mean?). If published, this will include your full peer review and any attached files.

Reviewer #2: No

---

## [Editor Report · Acceptance letter]

8 Aug 2023

PONE-D-23-02696R1 

Tissue-specific *in vivo* transformation of plasmid DNA in Neotropical tadpoles using electroporation 

Dear Dr. Delia:

I'm pleased to inform you that your manuscript has been deemed suitable for publication in PLOS ONE. Congratulations! Your manuscript is now with our production department. 

Kind regards, 

on behalf of

Dr. Michael Schubert 

Academic Editor

PLOS ONE